# Automated Observations of Dogs’ Resting Behaviour Patterns Using Artificial Intelligence and Their Similarity to Behavioural Observations

**DOI:** 10.3390/ani14071109

**Published:** 2024-04-04

**Authors:** Ivana Schork, Anna Zamansky, Nareed Farhat, Cristiano Schetini de Azevedo, Robert John Young

**Affiliations:** 1School of Sciences, Engineering & Environment, University of Salford, Manchester M5 4WT, UK; ivanaschork@gmail.com; 2Information Systems Department, University of Haifa, Haifa 31905, Israel; annazam@is.haifa.ac.il (A.Z.);; 3Department of Evolution, Biodiversity and Environment, Institute of Exact and Biological Sciences, Federal University of Ouro Preto, Ouro Preto 35402-136, Brazil; cristiano.azevedo@ufop.edu.br

**Keywords:** animal welfare, behavioural observations, computer vision, AI

## Abstract

**Simple Summary:**

Our research team has developed an automated computer system that uses convolutional neural networks (CNNs) to monitor and analyse the sleep patterns of dogs. Traditional methods of recording animal behaviour, such as direct observations (of sleep) of either live behaviour or recorded behaviour, can be time-consuming and error-prone, making it difficult to replicate studies. Sleep may be a crucial indicator of an animal’s well-being, but it has been overlooked in animal welfare research due to the time-consuming nature of measuring sleep. Compared to direct behavioural observations from the same videos, our system achieved an 89% similarity score in automatically detecting and quantifying sleep duration and fragmentation in dogs. Although there were no significant differences in the time percentage of sleep observed, the system recorded more total sleep time than human observers making direct observations on the same data sources. The automated system used could become a valuable tool for animal behaviour and welfare research.

**Abstract:**

Although direct behavioural observations are widely used, they are time-consuming, prone to error, require knowledge of the observed species, and depend on intra/inter-observer consistency. As a result, they pose challenges to the reliability and repeatability of studies. Automated video analysis is becoming popular for behavioural observations. Sleep is a biological metric that has the potential to become a reliable broad-spectrum metric that can indicate the quality of life and understanding sleep patterns can contribute to identifying and addressing potential welfare concerns, such as stress, discomfort, or health issues, thus promoting the overall welfare of animals; however, due to the laborious process of quantifying sleep patterns, it has been overlooked in animal welfare research. This study presents a system comparing convolutional neural networks (CNNs) with direct behavioural observation methods for the same data to detect and quantify dogs’ sleeping patterns. A total of 13,688 videos were used to develop and train the model to quantify sleep duration and sleep fragmentation in dogs. To evaluate its similarity to the direct behavioural observations made by a single human observer, 6000 previously unseen frames were used. The system successfully classified 5430 frames, scoring a similarity rate of 89% when compared to the manually recorded observations. There was no significant difference in the percentage of time observed between the system and the human observer (*p* > 0.05). However, a significant difference was found in total sleep time recorded, where the automated system captured more hours than the observer (*p* < 0.05). This highlights the potential of using a CNN-based system to study animal welfare and behaviour research.

## 1. Introduction

The study of animal welfare is often carried out through the measurement of animal behaviour. This is because an animal’s behaviour (i.e., welfare output) directly corresponds to its environmental conditions (i.e., welfare inputs) and the attempts of individuals to adapt [1]. Furthermore, behavioural observations are frequently preferred over other methods, such as physiological measures, due to their non-invasive nature and lower probability of interfering with individuals’ responses [2,3].

Despite being a simple method, behavioural observations (assessments) are not without their limitations. First, observing behaviour demands time from a human observer to quantify the behaviour. Second, a basic knowledge of the species’ behaviour is necessary to answer specific questions about welfare. Third, results are highly dependable on the reliability of scoring the same behaviours consistently over time, which demands the training of human observers to ensure inter-observer reliability if multiple observers are used. Lastly, not all animal-holding institutions allow researchers to observe animals outside their working/daytime hours, which causes a loss of important information over time; for example, zoo studies are biased towards daylight hours [3,4,5].

With advances in technology, it is possible to try to mitigate some of the problems of direct behavioural observations; for example, video monitoring of animals in different environments is used as an alternative to direct observations, but images still need to be quantified by a human observer [6,7]. Additionally, software has been developed to help score behaviour from videos to expedite the processing of images (e.g., [8]). Despite this, human-observed measurements of animal behaviour, even when computer-assisted, remain slow, labour-intensive, and prone to errors [4,6,9]. Such tedious and lengthy processes reduce the number of experiments that can be conducted, reduce the opportunity to work with larger sample sizes, and can limit the statistical power of the results [6,9].

Recent advancements in computer vision and deep learning could lead to the development of automated tracking and behaviour analysis systems that could revolutionise how behavioural variables are recorded. Examples of how these automated systems can help handle larger data sets can be found in the field of Ecology and Conservation. Using camera traps is a well-known and cost-effective methodology for monitoring populations without interference [10]. However, they also generate vast amounts of data from the photos and videos acquired during sampling [11,12]. Deep learning models have been used in all stages of data processing, from the system classification of photos with and without individuals to the categorisation of behavioural repertoires, thus providing an efficient tool for analysing large-scale camera-trap data [11,12,13].

Therefore, automated videos can increase scoring accuracy, replicability, and the number and nature of measured variables. This allows for the generation of larger data sets and more significant sample sizes, which increases statistical power [14,15,16]. These are the aims of the emergent field of computational behaviour analysis, also called computational ethology [6].

Automated video systems to record animal behaviour already exist for wild animals [17], farm animals [18], laboratory rodents [19], insects [20], and fish [21], and there are even well-developed commercial systems such as Ethovision [22]. Furthermore, in the field of animal welfare, such systems have been used to monitor pregnant cows before calving [23], aggression in pigs [24], and the activity of broiler chickens with different gait scores [25].

However, among the behaviours that automation has yet to explore is sleep, a well-researched indicator of humans’ good health and well-being [26], which has been mostly overlooked in animal welfare research.

The decades-long comprehensive study of the sleep/wake cycle, using both human and several non-human models, led to the conclusion that sleep is not a simple resting state but an essential physiological process that mediates individuals’ physiological and psychological functions and is an intrinsic part of the homeostatic process [27,28]. Moreover, the environment and events experienced during waking hours can affect the quantity and quality of sleep, and stress remains the main factor impairing sleep in both human and non-human animals [29,30,31,32]. As stress remains the primary source of stress in captivity [33], understanding sleep characteristics, especially sleep changes, is relevant to the health and well-being of animals under human care.

Still, the use of sleep as a metric is limited, most likely due to the difficulties in measuring such behaviour [34]. Not only is sleep behaviour challenging to measure due to the time-consuming and intensive nature of observations, but sleep has also been believed to provide accurate information about an animal’s biology only if assessed using EEGs.

Nonetheless, previous studies have attempted to quantify sleep and rest behaviour through video monitoring and this has proven to be an effective non-invasive technique. Some studies have scored over 90% confidence between observations and EEG, demonstrating the reliability of this method (e.g., [35,36]). Even though video monitoring is based on human observations, it still shows that measuring sleep using a video-based methodology is possible. Additionally, video monitoring is invaluable for such studies because cameras provide a spatial and temporal metric that can be used to assess most aspects of animals’ behaviour without interfering with the individuals [6,21].

Sleep also has desirable characteristics that may facilitate its use as a target behaviour for automated monitoring. Despite mammal species differing in some of the characteristics of their sleep patterns, such as the number of bouts and time of day, sleep still fulfils the same biological purposes. All species follow comparable sleep cycles, which commence with slow-wave sleep, followed by REM sleep (Rapid Eye Movement), and then wakefulness [37,38]. Moreover, virtually all mammals (except some marine mammals [39]) are immobile when sleeping or have sleep postures which can be linked to specific phases of their sleep cycle (e.g., horses [40] and cows [41]). Therefore, facilitating the assessment of sleep behaviour using automation should provide a valuable tool for animal welfare assessment.

In recent years, research has demonstrated that behavioural sleep information can provide valuable insights into an individual’s welfare (e.g., dogs [42], horses [40], giraffes [43], chickens [44]). Dogs, in particular, can be useful as models to study sleep welfare due to the unique relationship developed through coevolution with humans; dogs present certain cognitive and behavioural traits that enable them to have similar responses to the environment and towards other individuals, much more like humans than any other existent species [45]. Hence, they also make valuable models when studying complex subjects such as brain development, cognition, and sleep disorders (e.g., narcolepsy) [46,47]. Moreover, several characteristics make the domestic dog an ideal model species for animal welfare studies. First, they have well-known physiology and behaviour, including sleep parameters [48]. Second, dogs are accessible in large sample sizes and easily trained [49]. Third, pet dogs coexist with humans, which means they can provide information on how they cope with a world designed for humans [45,50,51].

Studies aiming to automate the investigation of dog behaviour have relied on wearable technology for pets to measure dog behaviour [52]. While these devices can measure activity and sleeping patterns, scientific validation is often lacking [53,54]. Furthermore, using this technology in clinical or scientific settings is not always appropriate.

Similarly, although some of these sensor-based activity trackers have achieved good accuracy [53,55,56], they are limited to a small number of behaviours (e.g., resting) and postures (e.g., lying down, sitting, etc.), which compromises their use in animal welfare assessment.

Despite being a well-studied species, only a few studies address the automatic video-based analysis of dog behaviour [14,57,58,59]. These studies automatically tracked individuals and detected dogs’ body parts using machine learning classifiers. However, the experiments used videos taken from 3D Microsoft Kinect cameras (Redmond, Washington, DC, USA) or street (security) surveillance systems whose installation is not trivial, and the devices are expensive.

In this study, we present a system that compares convolutional neural networks (CNNs) with direct human behavioural observation methods to detect and quantify dogs’ sleeping patterns using the same dataset. A CNN is a deep learning algorithm trained to classify behaviours directly from images. These networks learn how to use patterns to identify objects, faces, and scenes from image data [60]. Unlike previous studies using automatic video analysis, requiring specialised equipment, this system was designed to work on video footage obtained from low-quality, cheap, readily available cameras. It also has a user-friendly interface that produces a summary output of the variables measured, which removes the need for any advanced knowledge from the user to be able to use the system.

## 2. Materials and Methods

### 2.1. Ethical Statement

This study underwent review and approval by two ethics panels: the Science and Technology Research Ethics Panel at the University of Salford, Manchester (STR1617-80), and the Commission of Ethical Use of Animals in Research at the Universidade Federal de Ouro Preto, Minas Gerais, Brazil (Protocol 2017/04).

### 2.2. Video Acquisition of Dogs

For the study, cameras were installed in the kennel facilities of the University of Ouro Preto, Minas Gerais, Brazil. Thirteen mixed-breed adult dogs were observed during eight months in five-day recording periods, totalling 130 nights of recordings and 13,668 videos captured by the cameras. These were then used to develop the system at the Tech4Animals lab at the University of Haifa, Israel.

The videos were recorded on a domestic CCTV system (Swann SWDVK-845504, Santa Fe Springs, CA, USA) with night vision capability. The cameras were able to capture videos in two modes: full-colour mode, when the sun or a lamp illuminates the space, and grey-scale mode, during the night (dark) or with very low light levels, which triggers the infrared light and automatically switches the camera to night vision. Despite the camera’s HD resolution (1280 × 720), the video footage is technically considered low quality.

### 2.3. BlyzerDS System Overview

The BlyzerDS (Behaviour Analyzer—Dog Sleep) system is an extension of a previously developed system for automatic tracking of dog movement [57], which was adapted specifically for the needs of this project. The system is designed to take digital video footage as input and generate a summary of sleep parameters. It can calculate each dog’s amount of sleep in the footage and the number of sleeping bouts. The video is transmitted to the system server and processed frame by frame. The neural network performs two main tasks: marking the position of the dog or dogs and classifying their state as awake or asleep. The system accepts raw digital video as input and generates a summary of the sleep parameters for that video. The analysis is divided into two stages. In stage 1, the system application sends the raw digital video frame by frame to the server. The server application uses a RESNET neural network [61] that has been trained beforehand to detect dogs in a frame by outputting a bounding box around them (as shown in Figure 1). The frames containing the bounding box are then sent back to the system application to be processed in the second stage.

In stage 2, the system detects movement in a series of frames. If no movement is detected the dog is classified as asleep. The following steps are used to detect movement:The content of the bounding box is converted into black-and-white images.The image is blurred.The change (delta) between consecutive frames is calculated.The computed delta is binarized with a threshold.The binarized image is dilated to fill in the gaps.Contours are detected, and their area is computed.

After these steps, the system scores the dog as asleep or awake and returns a summary of sleep parameters for that video. Detailed information on the system architecture, including pre- and post-processing training algorithms, can be found in [62].

### 2.4. Training Data Set and System Evaluation

The system was trained using 80,000 frames extracted from the data set. The developers manually revised the output results to ensure each frame contained two attributes: a bounding box surrounding each identifiable dog and the dog’s state as awake or asleep. Any frames with unclear images of dogs or with no dogs were discarded from the analysis.

The system’s accuracy was evaluated using ten videos of 600 s each. The video set included videos with 0–2 dogs, day/night, and different dogs and kennels. The system processed the videos, and a testing set of 6000 frames annotated with the system’s predictions was manually checked for correctness by the developers.

### 2.5. Similarity of the System against Standard Behavioural Observations

To compare the system’s similarity to the manual recordings of behaviour, 15 random nights were selected to be evaluated by the system and a human observer. Behavioural observations were carried out using focal sampling with continuous recordings of behaviour [63]. Sleep duration was recorded at any time the animal was in a resting position, with eyes closed, and/or with no perceivable movement. Additionally, the number of sleeping bouts was recorded for the observed period. A sleeping bout was determined as the shifting between wakefulness, sleeping, and wakefulness, regardless of sleep duration. Data were summarised for the observed periods as the sleeping duration in seconds and the number of observed sleeping bouts. These are the metrics that were also summarised by the system output, and which were used to compare the similarity of methods. Behavioural data were scored in the software Boris v.7.0.12 (Behavioural Observation Research Interactive Software) [8].

### 2.6. Data Analysis

The summary of data from the automated system and the direct behavioural observations (summarized identically) were tested for normality using Anderson–Darling tests. All statistical tests were considered significant at *p* < 0.05. Results are presented as either total measured time or count totals. The system’s similarity was tested against the human-recorded observations of the same observation sessions using paired *t*-tests [64]. Statistical analyses were carried out using SPSS 27.0 [65].

## 3. Results

During the evaluation of the system using the training data set, the system correctly classified 5340 frames out of the 6000 tested, scoring an 89% accuracy. Of the 15 days submitted, three days had to be excluded from the final analysis since poor weather conditions caused the cameras to move and the lights to switch on and off, leading to an inaccurate analysis. An additional day was excluded due to the loss of three hours of footage, which prevented comparisons between the methods.

The automated system scored an average of sleep of 10.9 ± 2.2 h, against 9.7 ± 1.6 h recorded manually by human observers. Moreover, the system found an average of 15 ± 5 bouts per night, while the human observations returned 16 ± 3.5 bouts per night. The differences between the methods ranged from 0.13% to 2.68%, with a mean difference of 0.88% (Table 1).

There was a significant difference between the computer system and the manual recordings for the duration of sleep behaviour in seconds (t = 2.805, df = 10, *p* = 0.019). However, when the duration was converted to a percentage of time spent asleep versus awake, no statistical difference was found between the two methods of observation (*p* > 0.05). Similarly, no difference was found for the number of bouts recorded (*p* > 0.05).

## 4. Discussion

In this study, we have demonstrated that automating the monitoring of behaviours is possible and reliable. It, therefore, offers a practical solution for mitigating common problems associated with measuring animal behaviour.

Despite observing a significant difference in sleep duration, the system showed a high similarity score when classifying sleep behaviour. It was just as precise as a human when recording the number of bouts and percentage of time spent asleep. Additionally, the system was able to record more sleep than manual observations in some cases, which highlights the problems associated with manually going through thousands of hours of recordings. Human observers typically take up to three times longer than the actual video duration to score, which can cause fatigue, leading to measurement inconsistencies over time [6,9].

Using a system based on the recent advances in artificial intelligence (AI), it is possible to optimise data collection, increasing precision, replicability, and experimental throughput [7,14,16]. Furthermore, these systems could become a valuable instrument for monitoring behaviours that are laborious to assess, such as resting [15,16]. As AI continues to learn while it is being used it is possible that it could detect patterns of small changes of activity (e.g., tiny twitches) or find patterns in changes in body postures, which could be associated with the shifts between different sleep phases (i.e., from non-REM to REM). Integrating remote EEG detection and AI prediction systems with humans is accurate when detecting sleep stages (range 76–96% [66,67]). Additionally, AI algorithms can be developed to integrate data from multiple sources, such as sleep posture, heart rate variability, and electrocardiogram (ECG) signals, to provide a more comprehensive understanding of sleep stages and their physiological correlations [67,68]. In non-human species, investigations using remote EEG and sleep postures have been attempted in cows; however, posture was not a good predictor for light and deep NREM sleep, indicating that further research is still necessary and the technology needs further development to improve results [69].

The downside of using behavioural observations in our sleep research was the extensive hours of video monitoring necessary for the BlyzerDS System to provide an efficient and accurate solution. The system is still being developed to make it more accurate and precise in measuring sleep parameters. The problems associated with image processing were mostly related to image quality and ‘noise’ in the footage. For example, as the cameras are exposed to the weather, on windy days, they move significantly, blurring the images, which lead to inaccurate measuring. The poor lighting conditions at the kennels also made it challenging to identify the dogs if they were in darker corners. These conditions can be mitigated by adjusting the CNN algorithm to compensate for movement and light, although this would require several months of future work.

CNN-based systems such as the BlyzerDS system, with further improvements and new training data sets, can be modified to record more categories of behaviour and the behaviour of other species. This could lead to a system that can be universally used by people working with animals in many different environments.

## 5. Conclusions

The use of CNN-based systems such as the BlyzerDS system can potentially improve how we conduct research into animal behaviour. Even in its initial stages, the system showed good precision when evaluating the sleep behaviour of dogs. Developing an autonomous system for behaviour analysis may help mitigate common problems associated with processing large video data sets manually: an extremely time-consuming, tedious, and error-prone task.

## Figures and Tables

**Figure 1 animals-14-01109-f001:**
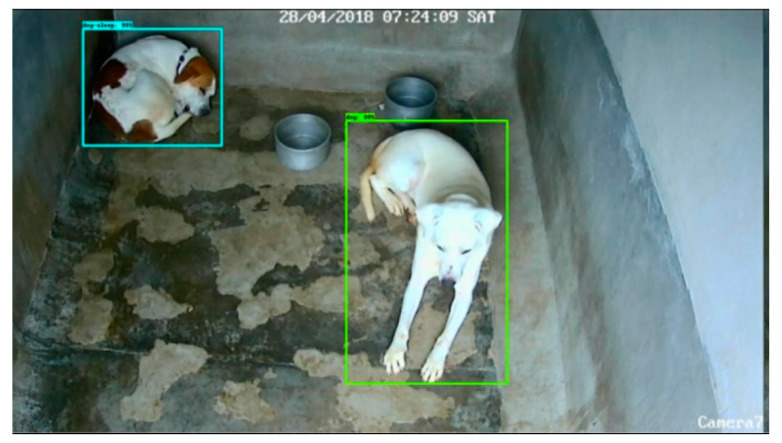
Detection of dogs by the BlyzerDS system using a neural network. Different colours in the bounding boxes show the system correctly scoring two individuals (blue = asleep; green = awake).

**Table 1 animals-14-01109-t001:** Summary of sleep metrics as recorded by the automated system (BlyzerDS-system) and compared with the recordings from manual (human) observations for 11 nights of behavioural observations. Sleep is represented in hours–minutes–seconds; bouts are reported as counts.

Sleep System	Sleep Manual	% Difference	Bouts System	Bouts Manual	% Difference
12:05:56	09:31:33	1.15	14	15	2.94
12:52:49	10:41:49	0.71	7	15	23.53
11:42:26	09:59:51	0.39	19	17	5.88
11:15:11	10:04:42	0.06	22	20	5.88
12:37:12	09:41:49	1.42	12	15	8.82
05:26:04	04:58:31	0.13	10	8	5.88
11:43:39	09:58:31	0.43	15	15	0.00
10:48:40	11:05:58	1.38	23	19	11.76
11:56:16	11:15:38	0.59	12	21	26.47
12:20:21	10:10:57	0.74	13	16	8.82
08:05:47	10:05:20	2.68	18	18	0.00

## Data Availability

Raw behavioural data used in the analysis is available online (Mendeley Data, V1, https://doi.org/10.17632/vfd7m2x38k.1 accessed on 1 June 2018).

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
