# Peer review of "Automated Observations of Dogs’ Resting Behaviour Patterns Using Artificial Intelligence and Their Similarity to Behavioural Observations"

_animals, 2024, doi:10.3390/ani14071109_

Round 1

Reviewer 1 Report

Comments and Suggestions for Authors

This manuscript detects the sleeping behaviour using CNN. CNN is a classic deep learning that can be used for object detection. Animals have some common daily behaviors, such as sleeping, eating, standing, etc. The detection of these behaviors plays an important role in the scientific breeding and management of animals. The comments are as follows,

1. The specific value of dog sleep behavior recognition for dog breeding and animal welfare, and the research purpose of the paper need to be further supplemented in the introduction and abstract.

2. Some achievements on animal behavior recognition should be cited and analyzed in the introduction.

I) A computer vision-based approach for behavior recognition of gestating sows fed different fiber levels during high ambient temperature.

ii) MammalClub: An Annotated Wild Mammal Dataset for Species Recognition, Individual Identification, and Behavior Recognition

iii) Pose estimation and behavior classification of broiler chickens based on deep neural networks.

Iv) Automatically identifying, counting, and describing wild animals in camera-trap images with deep learning

3. Representative example images in dateset should be showed.

4. One image is shown in Figure 1. In fact, the sleeping posture of animals is not always the same. You should add other sleeping postures in Figure 1.

Comments on the Quality of English Language

ok

Author Response

Please find the responses attached.

Reviewer 2 Report

Comments and Suggestions for Authors

The paper by Robert John Young and colleagues is an excellent work, which gives a nice introduction to automated (sleep) behaviour observation and then presents original results on a well-performing algorithm of dog sleep behaviour coding. I believe this will be an outstanding contribution to the Animals journal, and I highly recommend it for publication.

I very much enjoyed reading the introduction, which gives all the necessary literature background to automated behaviour observation. One thing that could additionally aid the reader might be to include in the text the species to which the authors refer to. There was some jumping back and forth between focusing on dogs, then giving again broader references (which is fine), but somewhat hard to follow – I had to repeatedly check reference numbers to see based on which articles the introductory claims are made.

I believe that the dataset presented is impressive and the methods / results are straightforward. One comment here: you find that your automated detection consistently (and for the duration variable significantly) overestimates sleep duration. I believe that the manual sleep duration data is already overestimated compared to what an EEG-based analysis would yield (video observation will falsely classify resting periods as sleeping). I still think that the ease of using the here presented method overrides these limitations, but nevertheless this is a serious drawback to consider, especially from a welfare perspective, as it might under detect true sleep problems.

I am also wondering if you can provide more details on how the detection performs with dogs of differing appearance. Were all N=13 dogs selected for this study white haired as on the photo? I expect that the system has problems with darker dogs considering that the kennel background is greyish. Can you please provide data about individual variance here. From the applied perspective it would be important to advise future users about the appearance of dogs for which the algorithm best works, and about others for which it is less reliable.

You briefly mention in the discussion that your algorithm might be able to detect (in the future) different sleep stages. This would be a very important next step, as sleep as such is not a uniform state, and poor welfare can result from superficial sleep (drowsiness) for the most time overtaking “proper” deep sleep (NREM and REM) during the sleep cycle. I believe that validating you system against non-invasive EEG data would be crucial here, can you please discuss this line more in detail?

Like I said, I believe that this paper is an extremely valuable contribution and the reliability numbers are highly convincing. Automated algorithm on dog sleep EEG data (or even manual coding of the same data) is not any more reliable, than what you achieved here (see e.g. Animals, 2020 10(6), 927.) Even if there is an alarming systematic bias, that your algorithm produces, you could argue with the relatively low inter-rater reliability of sleep EEG classification.

Author Response

Please find the responses attached.

Round 2

Reviewer 1 Report

Comments and Suggestions for Authors

Thank you for your work.

Comments on the Quality of English Language

OK